# Lignin-Degrading Enzymes and the Potential of *Pseudomonas putida* as a Cell Factory for Lignin Degradation and Valorization

**DOI:** 10.3390/microorganisms13040935

**Published:** 2025-04-18

**Authors:** Qing Zhou, Annabel Fransen, Han de Winde

**Affiliations:** Department of Molecular Biotechnology, Institute for Biology, Leiden University, 2333 BE Leiden, The Netherlands; q.zhou@biology.leidenuniv.nl (Q.Z.); fransenannabel@gmail.com (A.F.)

**Keywords:** lignin degradation, fungi, bacteria, enzyme, pseudomonas putida

## Abstract

Efficient utilization of lignin, a complex polymer in plant cell walls, is one of the key strategies for developing a green and sustainable bioeconomy. However, bioconversion of lignin poses a significant challenge due to its recalcitrant nature. Microorganisms, particularly fungi and bacteria, play a crucial role in lignin biodegradation, using various enzymatic pathways. Among bacteria, *Pseudomonas putida* is considered a promising host for lignin degradation and valorization, due to its robust and flexible metabolism and its tolerance to many noxious and toxic compounds. This review explores the various mechanisms of lignin breakdown by microorganisms, with a focus on *P. putida*’s metabolic versatility and genetic engineering potential. By leveraging advanced genetic tools and metabolic pathway optimization, *P. putida* can be engineered to efficiently convert lignin into valuable bioproducts, offering sustainable solutions for lignin valorization in industrial applications.

## 1. Introduction

Lignocellulose, composed of intertwined cellulose, hemicellulose, and lignin, is the most abundant renewable material on the Earth [1,2]. Lignin represents a class of complex and rigid organic polymers that form important structural and strengthening support in vascular plants and algae tissues. Lignin is the second most abundant terrestrial polymer on Earth after cellulose. Because of its stability and high recalcitrance, lignin has long been regarded as an industrial byproduct in pulp and paper waste, agricultural residues, and other hydrolytic industries [3,4].

The complex aromatic polymer is synthesized in plants mainly from three basic building blocks: *p*-coumaryl alcohol, coniferyl alcohol, and sinapyl alcohol. Consequently, the complex heterogenic polymeric lignin network mainly consists of three recognizable basic units: *p*-hydroxyphenyl (H), guaiacyl (G), and syringyl (S) moieties [5]. Gymnosperms (softwood, i.e., conifer, pine, cedar, spruce, etc.) mostly contain guaiacyl-based lignin (G), whereas dicotyledonous plants (legumes, beans, sunflower, tomato, etc.) mainly contain guaiacyl-syringyl-based lilac lignin (G-S), and monocotyledonous plants (maize, wheat, rice, cane, etc.) mainly contain guaiacyl-syringyl-hydroxyphenyl-based lilac lignin (G-S-H) [6] (Figure 1). Lignin polymerization in plants occurs via the formation of oxidative radicals of these structural units, followed by combinatorial radical coupling [7]. Among the variety of linkages, the β-O-4 bond is the most prominent one (40–50%), followed by β-β, β-5, 5-5, 5-O-4, and α-O-4 linkages [8,9,10,11] (for details, see Figure 1).

With the advent in recent years of biorefineries, millions of tons of lignin are available annually as a side product of industrial lignocellulose hydrolysis and utilization [12]. In the pulp and paper industry alone, around 100 million tons of lignin are becoming available as valuable but limited-use feedstock [13,14]. Cellulose and hemicellulose fractions are readily used as feedstock in subsequent biorefinery and fermentation scale-ups for the biotechnological production of various biofuels and biochemicals. In contrast, most lignin cannot be utilized efficiently. The utilization of lignin still largely resides in heat and energy production through combustion and the implementation of raw lignin in the production of glues, resins, and asphalt [15,16,17]. The global lignin market size was estimated at USD 1.08 billion in 2023 and is expected to grow at a compound annual growth rate (CAGR) of 4.5% from 2024 to 2030 [18]. Consequently, a more efficient and valuable utilization of lignin has recently gained growing interest.

Lignin degradation is important for the recycling and valorization of plant biomass and plays a crucial role in carbon cycling and nutrient cycling in ecosystems. The first and most important step in the valorization of lignin is the efficient decomposition and depolymerization of the complex and recalcitrant polymer. The chemical decomposition of lignin has been described [19,20,21]; however, it yields a complex and toxic mixture with a difficult-to-define composition. Such chemical hydrolysates are not very useful in subsequent added-value utilization [22]. The Organosolv process, effectively reducing lignin molar mass and heterogeneity was recently published as an interesting example [23,24,25]. However, obtained fractions still contain relatively undefined lignin polymers. Combined and integrated chemical and biochemical approaches may be much more favorable [26]. Biological decomposition by microorganisms may provide a promising route toward added-value utilization of lignin. Microbial enzymes may specifically target lignin for efficient degradation into aromatic monomers and oligomers. These can subsequently be converted into valuable biochemicals through microbial metabolic pathways, providing a sustainable route for lignin utilization in biotechnology.

White-rot fungi have been known for several decades to naturally degrade lignin, whereas brown-rot fungi are only capable of modifying the lignin network to a limited extent [27,28]. These microbes produce oxidative enzymes like laccases and various types of peroxidases, which play a significant role in the aspecific breakdown of lignin [29,30,31]. In addition, many bacteria have been reported to degrade lignin. Moreover, bacterial lignin degradation involves enzymatic cleavage that is geared toward specific lignin linkages, suggesting a targeted approach to lignin breakdown [30]. In this critical review, we summarize and discuss the current knowledge and understanding of lignin-degrading microorganisms, their enzymes, reported to be active in lignin degradation, as well as aspects of proposed catalytic mechanisms. Despite the large volume of literature on microbial lignin degradation existing to date, relatively little insight has been gained on enzyme specificity and catalytic mechanisms operating to yield defined degradation products. We propose and discuss the potential of the robust soil bacterium *Pseudomonas putida* as a suitable host and cell factory for industrial lignin valorization, genetic and enzyme engineering strategies to enhance the synthesis of value-added bioproducts derived from lignin, and subsequent industrial applications of these bioproducts.

## 2. Microbial Degradation of Lignin

### 2.1. Lignin Degradation by Fungi

Various pioneering studies have identified fungi as effective lignin-degrading microorganisms, secreting a variety of non-specific but relatively efficient lignin-decomposing enzymes. Typical fungi implicated with lignin degradation are listed in Table 1.

White-rot fungi are commonly associated with hardwood and are renowned for their high potency to degrade lignin [32]. Lignin degradation by white-rot fungi leads to the bleaching of wood, exposing cellulose and hemicellulose fibers. This degradation enables more efficient enzymatic hydrolysis of polysaccharides, facilitating the subsequent conversion of these fibers into fermentable sugars for bioethanol production [33]. This selective degradation makes white-rot fungi interesting for many biotechnological applications since they remove lignin and leave the valuable cellulose intact.

Brown-rot fungi typically grow on softwoods and constitute only 7% of wood-rotting basidiomycetes [34]. In contrast to white-rot fungi, they preferentially hydrolyze the cellulose component of lignocellulose while only partially oxidizing lignin. This involves Fenton oxidation chemistry, during which hydroxyl radicals are produced that may partly be independent of specific enzyme activity [35].

In 1984, the white-rot fungus *Phanerochaete chrysosporium*, was found to produce an extracellular lignin-degrading enzyme, an oxygenase, which catalyzes several oxidations in the alkyl side chains of lignin-related compounds [36]. Subsequently, *Phlebia radiata* was reported to degrade lignin, and three peroxidases and one laccase were purified and characterized from this fungus. These enzymes were shown to modify kraft lignin and phenolic compounds containing hydroxyl and methoxy groups [37]. In addition, *Pleurotus eryngii* was shown to remove lignin from cereal straw [38], and two isoenzymes of manganese peroxidase were purified from this fungus [39]. Several other white-rot fungi have been reported to degrade different types of lignin. The white-rot fungus *Trametes hirsuta* has been shown to secrete several laccases and peroxidases to degrade kraft lignin. The decrease in kraft lignin molecular weight is clearly correlated with the activities of these enzymes [40]. *Trametes versicolor* [41] and *Pycnoporus cinnabarius* operated through laccase as the major phenoloxidase [42].

**Table 1 microorganisms-13-00935-t001:** Lignin degradation by fungi: overview of strains, sources, and references. This table summarizes various fungi species known for their lignin-degrading capabilities. It includes the specific strains studied, the lignin sources they were tested on, the year of publication, and the corresponding references.

Microorganism	Strain	Lignin Source	Year	Ref.
White-Rot Fungi	*Phanerochaete chrysosporium* BKM-1767	Lignin model compounds	1984	[36]
White-Rot Fungi	*Phlebia radiata*	Kraft lignin	1988	[37]
White-Rot Fungi	*Pleurotus eryngii*	Cereal straw	1994	[38]
White-Rot Fungi	*Trametes versicolor*	Kraft lignin	1995	[41]
White-Rot Fungi	*Pycnoporus cinnabarius*	Pine wood	1996	[42]
White-Rot Fungi	*Ceriporiopsis subvermispora*	Pinus taeda Wood chips	2004	[43]
White-Rot Fungi	*Ganoderma lucidum* IBL-06	Lignocellulosic substrates	2010	[44]
White-Rot Fungi	*Phlebia* sp. MG-60	Oak wood	2012	[45]
White-Rot Fungi	*Dichomytus squalens*	Wheat straw Lignin	2013	[46]
White-Rot Fungi	*Pleurotus ostreatus*	Palm midrib	2018	[47]
White-Rot Fungi	*Trametes hirsuta*	Kraft lignin	2021	[40]
Brown-Rot Fungi	*Gloeophyllum trabeum* (*Lenzites trabea*) (Pers. ex Fr.) 83	Lignin model compounds	2008	[48]
Brown-Rot Fungi	*Postia placenta* MAD-698-R	Aspen (Modification of Lignin)	2009	[49]
Brown-Rot Fungi	*Fomitopsis pinicola*	Wheat straw Lignin	2013	[46]
Soft-Rot Fungi	*Aspergillus fumigatus*	Kraft lignin	1986	[50]
Soft-Rot Fungi	*Podospora anserina*	Wheat straw Lignin	2020	[51]
Fungi	*Aspergillus* sp.	Alkali lignin	2011	[52]

For the brown-rot fungus *Gloeophyllum trabeum*, a lignin degradation redox cycling process was proposed, involving two extracellularly produced quinones that reduce Fe^3+^ to Fe^2+^ [34]. This was supported by research on *Postia placenta* indicating up-regulation of genes associated with iron acquisition [53]. These Fe^3+^-reducing compounds play an important role since their low molecular weight enables them to access the cell wall structure in wood and initiate decay so that the larger lignin-degrading enzymes can access and act upon lignin [30]. Release of ^14^CO_2_ was observed when *Gloeophyllum trabeum* and *Postia placenta* were cultured with a non-phenolic, (O^14^CH_3_)-labeled lignin *β*-*O*-4 dimer model compound. Hence, these brown-rot fungi may produce enzymes that may specifically cleave the *β*-*O*-4 linkage in lignin (see Figure 1).

Apart from white-rot and brown-rot fungi, soft-rot fungi, such as *Aspergillus flavus*, *Aspergillus fumigatus*, and *Aspergillus* sp. LPB5, generally demonstrate only limited lignin degradation abilities and tend to be less efficient than other fungi proficient in lignin degradation [54]. However, *Aspergillus fumigatus* was reported to degrade kraft lignin through demethoxylation and dehydroxylation five times better compared to the white-rot fungus *C. versicolor* [50]. Moreover, the ascomycete *Podospora anserina* could cause 24% (*w*/*w*) of substantial lignin removal during the 7 days of growth [51], unambiguously confirming its ligninolytic activity.

### 2.2. Lignin Degradation by Bacteria

Several species of bacteria have been described to possess enzymes that can degrade lignin (Table 2). The following describes important examples of reported lignin-degrading bacterial species.

*Rhodococcus* spp.

Certain strains of *Rhodococcus* bacteria, such as *Rhodococcus jostii* RHA1 [55], are known for their lignin-degrading capacity. *R. jostii* RHA1 degrades lignin in lignocellulose as well as kraft lignin to a low-molecular-weight phenolic byproduct, as monitored by spectrophotometric assays [55]. *R. jostii* RHA1 encodes two putative so-called dye-decolorizing peroxidases, or DyP peroxidases (see below for description of enzymes). One was characterized as lignin peroxidase DypB, active in lignin breakdown [56]. A genetically modified *R. jostii* RHA1 was able to produce 330 mg/L 2,4-PDCA (pyridine-dicarboxylic acid) in 40 h from 1% wheat straw lignocellulose, corresponding to a conversion yield of approximately 16% of the available lignin fraction [57]. *Rhodococcus pyridinivorans* CCZU-B16 [58], isolated from soil, could under optimized conditions degrade 30.2% of alkali lignin (4 g/L) in 72 h.

*Bacillus* spp.

Bacteria of the genus *Bacillus* isolated from pulp and paper mill effluent exhibited the potential to degrade lignin [59]. For example, *Bacillus altitudinis* SL7 reduced lignin content by 44% when grown with alkali lignin [60]. *Bacillus pumilus* LSSC3 and *Bacillus atrophaeus* CL29 exhibited high oxidative laccase activity in kraft lignin degradation, measured by the oxidation of the lignin model compound guaiacol [61]. *Bacillus flexus* RMWW II showed lignin degradation by 20% at a lignin concentration of 400 mg L^−1^ [62]. *Bacillus ligniniphilus* L1 can utilize alkaline lignin as a sole carbon source, producing 15 types of aromatic compounds as identified via GC-MS analysis [63]. Transcriptomic data indicate at least four pathways putatively involved in lignin degradation and metabolization of breakdown products, including the Gentisate pathway, Benzoic acid pathway, and β-ketoadipate pathway. *Bacillus* sp. (CS-1 and CS-2) can degrade alkali lignin with high laccase activities detected in crude enzyme extracts [64]. Nevertheless, the specific lignin-degrading enzymes remain to be characterized.

*Pseudomonas* spp.

Several *Pseudomonas* strains, including *Pseudomonas putida* and *Pseudomonas fluorescens*, have been found to possess lignin-degrading enzymes. These bacteria are often investigated for their applications in bioremediation and lignocellulosic biomass conversion. *P. putida* A514 was able to grow with alkali-insoluble lignin as the sole carbon source [65]. Recently, *P. putida* NX-1, isolated from leaf mold samples, could grow on kraft lignin and was engineered for PHA production [66,67]. Genome analysis of *P. putida* NX-1 revealed putative enzymes involved in lignin decomposition, including dyp-type peroxidases, versatile peroxidases, manganese peroxidases, and laccases. However, their functions and contributions to lignin decomposition have not yet been experimentally characterized. The ability to catabolize a wide range of natural aromatinds [68,69] indicates that *P. putida* KT2440 holds potential to be an excellent host for lignin degradation. *P. putida* KT2440 could utilize alkaline pretreated liquor (APL), primarily composed of lignin, to produce mcl-PHA in relatively good yield under nitrogen depletion [22]. Furthermore, outer membrane vesicles (OMVs) from *P. putida* KT2440 have been implicated in the biodegradation of lignin-derived aromatic compounds [68,69]. The copper-dependent oxidase CopA from *P.* putida KT2440 was shown to be involved in extracellular lignin oxidation [70]. Moreover, *P. putida* was recently shown to produce cis,cis-muconic acid from PCA, which is an intermediate product of lignin degradation [71,72]. Hence, *P. putida* appears highly promising as a biotechnology host strain to produce valuable compounds from lignin [73].

*Streptomyces* spp.

Several actinobacterial species, such as members of the *Streptomyces genus*, have shown lignin-degrading potential. *Streptomyces viridosporus* T7A is an example of an actinobacterium with ligninolytic activity [74]. *Streptomyces* spp. F-6 and *Streptomyces* spp. F-7 can remove around 38% of lignin, after 12 days of culture [75]. Recently, *Streptomyces thermocarboxydus* DF3-3 was isolated for alkali lignin degradation [76], secreting ligninolytic enzymes, such as manganese peroxidase, laccase, and specific small laccases [77]. For this species, a total of seven lignin-based derivatives metabolic pathways were predicted: the *β*-ketoadipate pathway and peripheral reactions; the gentisate pathway; the anthranilate pathway; the homogentisic pathway; the catabolic pathway for resorcinol; the phenylacetate–CoA pathway; and the 2,3-dihydroxyphenylpropionic acid pathway [76]. *Streptomyces* sp. S6 isolated from a decaying oil palm empty fruit bunch can grow on kraft lignin as the sole carbon source. After 7 days of incubation with *Streptomyces* sp. S6, the loss of the molecular weight of kraft lignin was up to 55.3% [78].

*Sphingomonas* spp.

*Sphingomonas* species, and more specifically *Sphingomonas paucimobilis* SYK-6, have been shown to degrade lignin-related aromatic model compounds [79]. These bacteria are known for their ability to break down various lignin-related structures. SYK-6 was the first bacterium shown to harbor several functional lignin-degrading enzymatic routes, involving glutathione peroxidases and etherases (see below).


*Other Proteobacteria*


Proteobacteria like *Pandoraea* sp., *Enterobacter*, or *Ochrobactrum* have been confirmed can utilize lignin or lignin model compounds. *Pandoraea* sp. B-6 secreted extracellular ligninolytic enzymes to degrade kraft lignin [80]. The low-molecular-weight compounds of kraft lignin were detected by GC-MS. Proteomics suggested *Enterobacter lignolyticus* SCF1 was able to use lignin in both assimilatory and dissimilatory pathways [81]. *Ochrobactrum* was first reported to depolymerize and utilize lignin in 2018 [58].

**Table 2 microorganisms-13-00935-t002:** Lignin degradation by bacteria: overview of strains, sources, and references. This table summarizes various species of bacteria known for their lignin-degrading capabilities. It includes the specific strains studied, the lignin sources they were tested on, the year of publication, and the corresponding references.

Microorganism	Strain	Lignin Source	Year	Ref.
Actinobacteria	*Rhodococcus jostii* RHA1	Kraft lignin	2011	[56]
	* Rhodococcus erythropolis *	Alkali lignin	2012	[82]
	*Rhodococcus opacus* DSM 1069	Lignin	2013	[83]
	* Rhodococcus opacus * PD630	Alkali Corn Stover Lignin	2017	[84]
	* Rhodococcus pyridinivorans * CCZU-B16	Alkali lignin	2018	[58]
	*Amycolatopsis* sp. 75iv2	Acid-precipitable, polyphenolic, polymeric lignin (APPL)	2011	[85]
	*Streptomyces viridosporus* T7A	APPL	1983	[74]
	*Streptomyces* spp. F-6	Alkali lignin	2012	[75]
	*Streptomyces* spp. F-7	Alkali lignin	2012	[75]
	*Streptomyces coelicolor* A3(2)	Lignin model compounds	2014	[86]
	*Streptomyces* sp. S6	Kraft lignin	2020	[78]
	*Streptomyces thermocarboxydus* DF3-3	Alkali lignin	2022	[76]
	*Micromonospora* sp.	Kenaf	2014	[87]
	* Thermobifida fusca * YX	Untreated biomass	2011	[88]
Anaerobic Microorganisms	*Clostridium thermocellum*	Populus Lignin	2017	[89]
Brevibacillus	*Brevibacillus thermoruber*	Lignin	2021	[90]
	* Caldicellulosiruptor bescii * DSM 6725	Untreated switchgrass	2013	[91]
Bacteroidetes	*Sphingobacterium* sp. HY-H	Sodium lignosulfonate	2013	[92]
	*Sphingobacterium* sp. T2	Wheat straw Organosolv lignin, alkali kraft lignin	2015	[93]
	*Sphingomonas paucimobilis* SYK-6	dimeric lignin compounds	1999	[79]
Proteobacteria	*Citrobacter* sp. (HQ873619)	Black liquor	2011	[94]
	*Citrobacter* sp. (FJ581023)	Black liquor	2011	[95]
	*Citrobacter freundii* (FJ581026)	Black liquor	2011	[95]
	*Comamonas* sp. B-9	Kraft lignin	2012	[96]
	*Comamonas testosterone* KF-1	Lignin-associated monomers	2023	[97]
	*Klebsiella pneumoniae* (GU193983)	Black liquor	2011	[94]
	*Klebsiella pneumoniae* NX-1	Kraft lignin	2018	[66]
	*Pseudomonas aeruginosa* (DSMZ 03504)	Pulp mill effluents	2010	[98]
	*Pseudochrobactrum glaciale*	Pulp paper mill effluent	2012	[99]
	*Pantoea* sp.	Pulp paper mill effluent	2012	[99]
	*Pseudomonas putida* KT2440	Alkaline pretreated liquor	2014	[22]
	*Pseudomonas plecoglossicida* ETLB-3	Black liquor	2015	[100]
	*Pseudomonas putida* A514	Alkali lignin	2016	[65]
	*Pseudomonas strain*	Alkaline insoluble lignin	2016	[65]
	*Pseudomonas* sp. Q18	Alkali lignin	2018	[101]
	*Pseudomonas putida* NX-1	Kraft lignin	2018	[66]
	*Pseudomonas strain* Hu109A	Lignin	2023	[102]
	* Pandoraea * sp. B-6	Kraft lignin	2013	[80]
	* Enterobacter soil * sp. *nov.*	Alkali lignin	2011	[103]
	* Enterobacter lignolyticus * SCF1	Alkali lignin	2013	[81]
	* Ochrobactrum pseudogrignonense *	Nitrated lignin	2012	[82]
	* Ochrobactrum rhizosphaerae *	Nitrated lignin	2012	[82]
	*Ochrobactrum tritici *NX-1	Kraft lignin	2018	[66]
	*Serratia marcescens* (GU193982)	Black liquor	2011	[94]
	*Serratia liquefaciens*	Pulp paper mill effluent	2012	[99]
	*Serratia liquefaciens* LD-5	Pulp paper mill effluent	2016	[104]
Firmicutes	*Aneurinibacillus aneurinilyticus* (AY856831)	Kraft lignin	2007	[59]
	*Bacillus* sp. (AY952465)	Kraft lignin	2007	[59]
	*Bacillus* sp. (accession no. AY 952465)	Kraft lignin	2007	[105]
	*Bacillus cereus* (DQ002384)	Kraft lignin	2008	[106]
	*Bacillus atrophaeus* LSSC3	Kraft lignin	2013	[61]
	*Bacillus pumilus* CL29	Kraft lignin	2013	[61]
	*Bacillus* sp. (CS-1 and CS-2)	Alkali lignin	2014	[64]
	*Bacillus megaterium* ETLB-1	Black liquor	2015	[100]
	*Bacillus ligniniphilus* L1	Alkali lignin	2017	[107]
	*Bacillus endophyticus*	Lignin	2016	[108]
	*Bacillus subtilis*	Lignin	2016	[108]
	*Bacillus flexus* RMWW II	Alkali lignin	2019	[62]
	*Bacillus altitudinis* SL7	Purified synthetic alkali lignin	2021	[60]
	* Paenibacillus * sp. (AY952466)	Kraft lignin	2008	[106]
	* Paenibacillus glucanolyticus * SLM1	Biochoice lignin	2016	[109]
	* Paenibacillus glucanolyticus * 5162	Biochoice lignin	2016	[109]
	* Paenibacillus * sp. *strain* LD1	Kraft lignin	2014	[110]
	*Planococcus* sp. TRC1	Lignin	2019	[111]
Extremophile bacteria	*Arthrobacter* sp. *C2*	Sodium lignin sulfonate	2022	[112]

## 3. Enzymes for Lignin Depolymerization

Several successful examples of *P. putida* converting lignin-derived compounds into valuable products have been reported [113]. However, despite these promising findings, achieving conversion starting from intact lignin remains challenging. Lignin degradation is a complex process that requires multiple enzymes and pathways. Many research efforts have been employed, trying to uncover the intricacies of these processes. These have yielded insights into fungal and bacterial enzymes with activity toward lignin degradation, with various bacterial enzymes putatively operating with higher specificity toward different lignin-specific linkages.

The initial and crucial step to effectively degrade lignin is to attack and depolymerize the complex lignin polymeric network into smaller phenoxy radical intermediates [114]. This step can be facilitated by external oxidoreductases, including laccase (Lac, EC 1.10.3.2), lignin peroxidase (LiP, EC 1.11.1.14), manganese peroxidase (MnP, EC 1.11.1.13), dye-decolorizing peroxidases (Dyp, EC 1.11.1.19), and versatile peroxidase (VP, EC 1.11.1.16) [115]. Typical enzymes capable of cleaving specific lignin linkages are summarized in Table 3. These enzymes have been extensively studied for their activity, however, almost exclusively on lignin model compounds [11]. They are known to target various linkages that occur within the lignin structure. These enzymes exhibit different substrate specificities and mechanisms of action; however, in most cases, their precise role in bioconversion of the lignin polymeric network remains elusive.

### 3.1. The β-O-4 Bond

A number of enzymes are secreted by fungi and bacteria to degrade lignin or lignin-derived compounds [129]; however, there is limited evidence regarding their ability to cleave specific linkages within the complex lignin structure. Reported evidence for linkage specificity mostly stems from studies with relatively simple model compounds for each of the lignin linkages [11]. Among the various linkages present in lignin, 45–60% of the total linkages are *β*-O-4 aryl ether bonds [130]. Cleaving this bond presents an essential step in the efficient use of lignin. Hence, enzymes that can cleave *β*-O-4 aryl-ether bonds are highly interesting for application in lignin valorization.

#### 3.1.1. Fungal Lignin Depolymerization Enzymes

Lignin Peroxidases (LiPs)

Lignin peroxidase, a monomeric heme-containing enzyme, was the first enzyme found in *P. chrysosporium* that can degrade lignin [27,131]. Its proficiency lies in the effective degradation of non-phenolic lignin units by catalyzing oxidative breakdown in the presence of H_2_O_2_ [132]. Therefore, it can catalyze the cleavage of *β*-*O*-4 ether bonds and Cα-C*β* linkages. LiPs are considered strong biocatalyst in the bioremediation of lignin and are represented in *Phanaerochaete chrysosporium*, *Trametes versicolor*, *Phanaerochaete sordida*, and *Phlebia radiata*.

Laccases

Laccases are widely found in plants, insects, fungi, and bacteria [133,134,135]. As a copper-containing enzyme of the polyphenol oxidases group, laccase catalyzes the oxidation of aromatic compounds, including phenols and phenolic derivatives during lignin degradation. Oxidation of these phenolic compounds leads to the formation of phenoxyl radicals, resulting in subsequent hydrolysis of C-C and β-aryl bonds in lignin’s aromatic rings [33], yielding various products such as syringaldehyde, 1-(3,5-dimethoxy-4-ethoxyphenyl)-2-hydroxyethanone, 1-(3,5-dimethoxy-4-ethoxyphenyl)-2-hydroxypropanal, and 2,6-dimethoxy-*p*-benzoquinone. Laccases are present in various fungi species such as *Dichomitus squalens*, *Irpex lacteus*, *Lentinula edodes*, *Cerrena maxima*, *Trametes versicolor*, *Pleurotus ostreatus*, and *Phanaerochaete chrysosporium* [132]. Interestingly, *Peniophora lycii* LE-BIN 2142 lacks ligninolytic peroxidases, which are typically considered key enzymes in white-rot fungi. Instead, this species primarily relies on multiple laccase isozymes and unique FAD-binding proteins, suggesting an alternative oxidative strategy for lignin modification [136].

Versatile Peroxidases (VPs)

Versatile Peroxidase, a heme-containing ligninolytic peroxidase, was first found in white-rot fungi *Pleurotus eryngii* [137]. VPs have been characterized to have catalytic functions of LiP, capable of oxidizing high redox potential substrates, combined with MnP, which oxidizes Mn^2+^ to Mn^3+^, producing a diffusible oxidizing agent effective on low redox potential species. In the absence of mediators, they also oxidize azo-dyes and other non-phenolic compounds with high redox potentials [138]. Different from MnPs and LiPs, VPs have a wider range of substrates. Evidence shows that VPs could catalyze *β*-*O*-4 lignin dimer to monomeric products [139]. Additionally, the VPs from *Physisporinus vitreus* oxidized the *β*-*O*-4 dimer, guaiacylglycerol *β*-guaiacyl ether, by depolymerization to a monomer or polymerization to a tetramer concurrently [119]. VPs are found in *Pleurotus, Bjerkandera* sp., *Panus*, *Calocybe*, *Trametes*, *Lepista*, *Dichomitous*, and *spongipelli* fungi species [33].

Manganese Peroxidases (MnPs)

Mangenese Peroxidase catalyzes the oxidation of a non-phenolic aromatic ring structure in lignin via oxidation of Mn^2+^ to Mn^3+^ as a redox mediator, leading to structural cleavage [140]. MnP from *Phanerochaete chrysosporium* was found able to cleave the *β*-*O*-4 of the phenolic lignin model dimer 1-(3,5-dimethoxy-4-hydroxyphenyl)-2-[4-(hydroxymethyl)-2-methoxyphenoxy]-1,3-dihydro-xypropane [120,141]. MnP was first discovered in *P. chrysosporium* but was also later detected in other Basidiomycota species, including *Panus tigrinus*, *Lenzites betulinus*, *Agaricus bisporus*, *Bjerkandera* sp., and *Nematoloma frowardii* [138].

Dye-decolorizing Peroxidases (DyPs)

Lastly, dye-decolorizing peroxidases (DyPs) are evolutionarily not related to the classical LME peroxidases (LiPs, MnPs, and VPs) but are a new class of heme-containing peroxidases found in bacteria and fungi [33]. DyPs were first isolated in 1999 from the basidiomycetous fungus *Bjerkandera adusta* [142]. Some ligninolytic activity was found in *Termitomyces albuminosus*, *Auricularia auricula-judae*, and *Irpex lacteus.*

#### 3.1.2. Bacterial Lignin Depolymerization Enzymes

In addition to harboring enzymes with characteristics comparable to fungal lignin-degrading enzymes, bacteria also have different lignin degradation mechanisms and enzymes. In the 1980s, *Pseudomonas acidovorans* had already been reported to degrade a *β*-aryl ether model compound [143]. Nevertheless, as of today, the actual number of functional bacterial enzymes well characterized in detail remains limited.

*β*-Etherase

*β*-Etherase, belonging to the protein superfamily of glutathione-S-transferase (GST; EC 2.5.1.18), is the first bacterial gene reported to function specifically in lignin degradation in *Sphingobium* sp. SYK-6 [144,145,146]. *β*-Etherases exist especially in microorganisms that specialize in decomposing lignin [147]. The *β*-*O*-4 aryl-ether bond degradation pathway in *Sphingobium* sp. SYK-6 needs three steps, involving three enzymes: an NAD^+^-dependent Cα-dehydrogenase (LigD, LigL), a *β*-Etherase (LigE, LigF), and LigG, a glutathione-dependent lyase (LigG) [148,149]. First, LigD/LigL oxidizes the Cα in model substrates, like 1-(4-hydroxy-3-methoxyphenyl)-2-(2-methoxyphenyl) propane-1,3-diol (GGE), under consumption of NAD^+^. Only after this oxidation, LigE or LigF can cleave the C*β* ether bond, following the S_N_2-type mechanism with high stereoselectivity. While LigE cleaves ether bonds in substrates with (R)-configured *β*-carbon, resulting in the corresponding (S)-configured glutathione adducts, LigF converts the corresponding (S)-substrate enantiomers [150]. Finally, LigG catalyzes the thioether cleavage of the chiral glutathione adducts to produce oxidized glutathione (GSSG) [151,152], as shown in Figure 2.

Dye-decolorizing Peroxidases (DyPs)

Dyps, heme-containing peroxidases are regarded as important enzymes involved in lignin degradation, since they can specifically cleave and degrade a list of such lignin model dye compounds [153]. Generally, peroxidase can catalyze the degradation reaction of hydrogen peroxide, leading to the generation of reactive oxygen species, which in turn participate in lignin degradation. Additionally, DyP enzymes also catalyze the oxidation of *β*-*O*-4 linkages, converting veratrylglycerol-*β*-guaiacyl ether into veratryl aldehyde and cleaving guaiacylglycerol-*β*-guaiacyl ether [154,155]. The DypB from *Rhodococcus jostii* RHA1 was the first bacterial lignin-degrading enzyme that has been characterized, which is capable of oxidizing polymeric lignin and lignin model compounds [56]. Novel research also found that Dyp1B from *Pseudomonas fluorescens* plays a significant role in lignin degradation [156].

Laccase-like multicopper oxidases (LMCOs)

Laccase-like multicopper oxidases (LMCOs) are a diverse group of oxidoreductases found in bacteria, fungi, and plants [157]. CopA is a member of LMCOs or pseudo-laccases [158,159]. CopA enzymes from *P. putida* KT2440 and *P. fluorescens* Pf-5 catalyze the oxidization of the lignin model compound GGE (see above, Figure 2) and 2,2′-dihydroxy-3,3′-dimethoxy-5,5′-dicarboxybiphenyl (DDVA, see blew, Figure 3), producing oxidized dimerized products [70,160].

Laccases

Laccase plays a crucial role in lignin biodepolymerization, but the reaction mechanism in bacteria remains incompletely elucidated.

Among bacterial laccases, small laccases (SLACs) are a type of laccase enzyme characterized by their smaller molecular size compared to traditional laccases [161]. The SLAC from *Streptomyces* can degrade a phenolic *β*-*O*-4 lignin model compound (LM-OH) [86]. Furthermore, SLAC variants have been functionally expressed in *Aspergillus niger* and are active in lignocellulose degradation [77].

The laccase from *Bacillus ligniniphilus* L1 was found to promote lignin degradation by oxidizing phenolic and non-phenolic structures in lignin [121]. In addition, this study highlights its potential role in cleaving key interunit linkages in lignin, including β-O-4, β-5, β-β, 4-O-5, and 5-5.

#### 3.1.3. 5-5 Bond (Biphenyl Bond)

The proportion of 5-5 bonds in lignin is around 10% in softwood and 5% in hardwood [162]. Remarkably, it has been demonstrated that the cleavage of the biphenyl linkage plays a pivotal role in facilitating lignin degradation.

Amongst fungi, the versatile peroxidases (VPs) in *Physisporinus vitreus*, were also observed to cleave the 5-5 bond of dehydrodivanillic alcohol (5-5′ dimer) in vitro for the first time [119].

The bacterial biphenyl degradation pathway was also found in *Sphingobium* sp. SYK-6 by growing on 2,2′-dihydroxy-3,3′-dimethoxy-5,5′-dicarboxybiphenyl (DDVA) [163]. In the mechanism of degrading DDVA, four enzymes are involved: LigX (a non-heme iron-dependent demethylase), LigZ (an extradiol dioxygenase), LigY (a C-C hydrolase), and LigW/LigW2 (decarboxylases) [122,123,124]. In the catalytic progression of DDVA, the enzyme LigX catalyzes the elimination of a methoxy group, resulting in the formation of a hydroxyl group. Subsequently, the product generated by LigX serves as a substrate for oxidative meta-cleavage, facilitated by LigZ. Following this, LigY transforms the ring fission product into 4-carboxy-2-hydroxypentadienoic acid and 5-carboxyvanillic acid (5CVA). This sequence culminates with the participation of LigW and LigW2, which convert 5CVA into the pivotal metabolic intermediate, vanillic acid or vanillate, essential for the synthesis of various bioproducts, as shown in Figure 3.

#### 3.1.4. β-β Bond (Resinol Bond)

The breakdown of the pinoresinol lignin model compound has also been studied in *Fusarium solani* M-13-1 and *S. paucimobilis* SYK-6 [124,125]. The catabolic pathways for both heterocyclic lignin components appear to involve alpha-hydroxylation as an initial step. However, enzymes that participate in the reactions have not been characterized clearly. Until 2018, the isolation of the highly efficient (+)-pinoresinol-mineralizing *Pseudomonas* sp. strain SG-MS2 and its catabolic pathway were reported, highlighting a significant advancement in understanding the catabolism of pinoresinol lignin dimers, as shown in Figure 4 [164].

#### 3.1.5. Other C-C Bonds

Fungal MnP possesses the capability to not only break *β*-*O*-4 linkages in phenolic structures but also disrupt C*α*-C*β* and *β*-aryl ether bonds in non-phenolic substances [165,166]. Studies indicate that laccases could cleave Cα−Cβ bonds or aryl−Cα bonds and catalyze the oxidation of Cα−OH to Cα=O of lignin model compounds [167]. The laccase degradation mechanism may vary depending on the substrate, pH, temperature, and other environmental conditions [168]. In addition, different types of laccases may have different substrate specificity and degradation efficiency. A deep understanding of these mechanisms is needed for developing effective biotechnological applications for lignin degradation.

## 4. *Pseudomonas putida* as a Lignin-Degrading Cell Factory

Although the efficiency of lignin depolymerization by bacterial extracellular enzymes is less well studied than that of white-rot fungi, bacteria do provide a flexible platform for the heterologous expression of ligninolytic enzymes [169]. We and others have investigated endogenous lignin degradation with strains of *Pseudomonas putida*. Moreover, strains of *P. putida* have been implicated in bioconversion and biosynthesis of valuable products from lignin-derived compounds [71,72,170]. The direct and grand challenge now is to directly access and utilize lignin as a source of valuable products. *P. putida* may prove a promising vehicle for the expression of various specific ligninolytic enzymes to construct cell factories, enabling the direct conversion of lignin into high-value products (see Figure 5).

### 4.1. Natural Capabilities and Metabolic Pathways

*P. putida* strains like S12 and KT2440 are recognized as highly promising industrial host strains [171]. These strains can adapt to diverse physiochemical and nutritional niches and possess robust metabolic redox power, enabling them to survive under high oxidative stress. Moreover, *P. putida* has been found in natural environments degrading various organic compounds, including lignin-derived molecules. For example, *P. putida* KT2440 has been identified to possess the ability to degrade p-hydroxybenzoate, benzene, and xylene, which are components of lignin-derived aromatic hydrocarbons [172]. *P. putida* has a wide range of metabolic functions, combined with its extensive catabolic pathways, enabling it to utilize lignin-derived aromatic compounds as carbon sources.

In recent years, the depolymerization of lignin has become a research hotspot [173,174]. Lignin-derived compounds like ferulic acid and vanillin have been given particular attention [175]. *P. putida* KT2440 can metabolize vanillin by conversion into vanillate, at a rate of 4.87 mmol (gCDW _h_)^−1^ [176]. Furthermore, *P. putida* KT2440 can degrade ferulic acid via a CoA-dependent non-β-oxidative pathway [177]. Additionally, this strain can metabolize benzoate and catechol through the native β-ketoadipate pathway, further demonstrating its ability to process lignin-derived compounds [172]. The β-ketoadipate pathway is a chromosomally encoded aromatic compound degradation pathway that is widespread among soil bacteria and fungi [178]. Such pathways are essential for lignin degradation and valorization, turning complex aromatic polymers into economically valuable products.

*P. putida* demonstrates exceptional performance in the degradation of lignin derivatives and other aromatic compounds, showing the potential applications of biotechnology in lignin valorization. Therefore, *P. putida* strains are promising suitable platforms for the bioconversion of exogenous toxic chemical streams into valuable products that derive from lignin degradation.

### 4.2. Genetic Engineering of Pseudomonas putida

#### 4.2.1. Genomic Tools

To develop efficient cell factories for lignin degradation, a specifically robust bacterial chassis is needed. *P. putida* is a robust platform with advanced metabolic engineering applications [179]. The degradation capability of *P. putida* toward lignin can be further enhanced through genetic engineering. The introduction of genes encoding ligninolytic enzymes from other microorganisms can improve efficiency. Modifying the metabolic pathways of *P. putida* can convert the lignin degradation products into valuable compounds.

Over the years, dedicated genetic tools have been developed for the expression or deletion of genes in *P. putida* [180,181]; see Table 4. With the use of gene editing tools, it is highly possible to improve the lignin degradation of *P. putida*. For example, previous studies show that the expression of *Pseudomonas fluorescens* Dyp1B in *P. putida* KT2440 results in enhanced activity for the oxidation of 2,6-dichlorophenol (DCP) and polymeric lignin [156].

#### 4.2.2. Secretion System

Lignin, as a polymeric network, is obviously too large and complex to be transported into the cell and be degraded intracellularly [191]. Hence, lignin-degrading enzymes for biotechnological applications should be extracellularly secreted with the microbial secretome. Several secretion systems have been studied in *P. putida*: Outer membrane vesicles (OMVs) are secreted by the bacterium to deliver enzymes [192]. Pioneering research has shown that OMVs in *P. putida* KT2440 can catabolize lignin-derived aromatic compounds [69]. This property can be exploited to deliver ligninolytic enzymes to lignin substrates, thereby enhancing the degradation process. A novel recombinant peroxidase secretion system has been constructed in *P. putida* KT-M2 [193]. A flagellar type III secretion system was used for the dye decolorization peroxidase of *P. putida*, resulting in efficient oxidative activity of cell-free supernatants against a variety of chemicals, including the lignin model compound. Additionally, the periplasmic expression of peroxidase Dyp1B has been explored for lignin valorization in *P. putida* [156]. The periplasmic expression strain shows higher lignin oxidation activity than the wide type.

These advancements in secretion systems, together with the genetic engineering tools, highlight *P. putida*’s potential as a powerful biotechnological platform for lignin degradation. By combining these approaches, it is possible to enhance the efficiency of lignin valorization processes, turning this complex and recalcitrant polymer into valuable bioproducts.

### 4.3. Biological Conversion and Reutilization

Biological funneling is a concept in bioconversion and metabolic engineering where a diverse array of complex molecules is funneled through a series of biological pathways to produce a single or a few specific valuable products [194,195]. Biological funnels can overcome the challenging heterogeneity of chemical mixtures as recently applied and demonstrated with low-molecular-weight lignin-derived aromatics [196,197]. Indeed, *P. putida* may convert a mixture of lignin degradation products into useful compounds. These compounds can be further utilized, such as biofuels, chemical raw materials, and other bio-based products.

To obtain low-molecular-weight lignin-derived molecules, we may rely on a combination of chemical and biological treatment. Recently, conversion from lignin to medium chain-length polyhydroxyalkanoates was achieved in *P. putida* by combining microbial treatment with chemical pretreatment [22]. In a comparable effort, lignin conversion to β-ketoadipate was achieved with engineered *P. putida* [198]. In this work, genes encoding enzymes mediating 4-hydroxybenzoate hydroxylation and vanillate O-demethylation were overexpressed to improve the yield, and the gene that could cause intermediate accumulation was deleted. Additionally, through genetic engineering, Altenbuchner et al. successfully introduced key enzymes involved in the conversion of lignin-derived ferulic acid to vanillin in *P. putida* KT2440 [170]. This research enhances vanillin production with up to 86% molar yields and few byproducts.

These studies further illustrate that engineering metabolic pathways in *P. putida* to funnel lignin breakdown products into desirable bioproducts can increase the economic value of lignin valorization processes.

## 5. Conclusions and Future Prospects

Lignin, the second most abundant terrestrial polymer found on Earth, constitutes an important part of plant fibers [199]. Lignin is composed of a network of aromatic compounds and is highly resistant to decomposition. Currently, this rich aromatic compounds resource is mainly separated as a waste stream, where 98% is used as a heat source in factories [200]. Only 2% is used in a chemical conversion to produce useful compounds like lignosulfonates [17].

Importantly, we here make a case for lignin to be utilized much more effectively through biotechnological and chemical processes. Using those techniques, separately or in combination, lignin can be degraded into hundreds of valuable derivatives [201]. In nature, bacteria and fungi can degrade lignin, with some differences in the degradation mechanism, substrate specificity, and product generation. Only recently, research on the enzymatic processes involved in bacterial lignin degradation has led to the identification and documentation of specific enzymes dedicated to this purpose. This review provides a list of microorganisms reported to utilize lignin and potential enzymes involved in specific lignin depolymerization.

Fungi, especially white-rot fungi, produce a variety of enzymes (such as lignin peroxidase, manganese peroxidase, laccase, etc.) that can directly oxidize and degrade lignin. Fungi typically work by producing multiple enzymes that work together to break down different lignin bonds and connections. The depolymerization of native lignin is facilitated by extracellular oxidative enzymes, including Lip, MnP, VP, and Lac, which have been extensively documented in fungi. The research on bacterial degradation of lignin is not as in-depth as that on fungi. It has only been a dozen years since the first bacterial enzyme that degrades lignin was discovered [56]. In bacteria, the lignin degradation enzyme systems are thought to be relatively simple and specific. Limited enzymes are involved, such as phenol oxidase. In bacteria like *Cupriavidus basilensis* B-8, Lac, and MnP activities were identified; however, no MnP or Lac genes were found [202]. Hence, bacteria are anticipated to possess distinctive lignin degradation mechanisms and novel types of peroxidases [169].

Typically, fungal lignin degradation spans 10–30 days, whereas in bacteria, it may be accomplished in as little as 2–7 days. From an industrial viewpoint, utilizing bacteria as a host strain to establish a lignin degradation and utilization cell factory would prove more cost-effective [203]. In bioreactors, bacteria have advantages over fungi due to their rapid growth, simpler cultivation requirements, higher metabolic rates, easier genetic manipulation, and simpler product recovery [204,205]. While research on the enzymatic processes of lignin-degrading bacteria is currently relatively limited, using bacteria for lignin degradation still holds great prospects. This review demonstrates the great potential of *P. putida* as a microbial cell factory in lignin degradation and valorization, providing a sustainable approach to converting lignin into valuable bioproducts. The plasmid-free strain *P. putida* KT2440 is particularly regarded as a microbial host for biotechnological applications due to its biosafety status [206] and is widely used in industrial production.

We believe that, through genetic engineering and process optimization, *P. putida* can be adapted to industrial needs and contribute to further developing the bioeconomy through sustainable industrial practices.

## Figures and Tables

**Figure 1 microorganisms-13-00935-f001:**
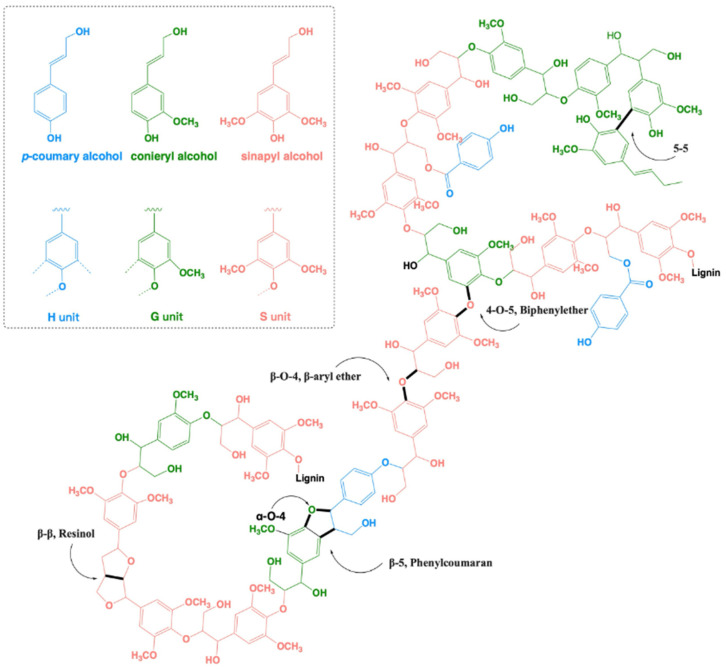
A structural representation of the complex lignin polymeric network (adapted from [5,7]). The three basic units constituting lignin are *p*-coumaryl alcohol (blue), coniferyl alcohol (green), and sinapyl alcohol (red). These monomeric alcohols are linked to form lignin mainly by the following linkages: β-O-4 (β-aryl ether) linkages, *β*-*β* (resinol), *β*-5 (phenylcoumaran), 5-5 (biphenyl), and 4-O-5 (biphenylether) bonds.

**Figure 2 microorganisms-13-00935-f002:**
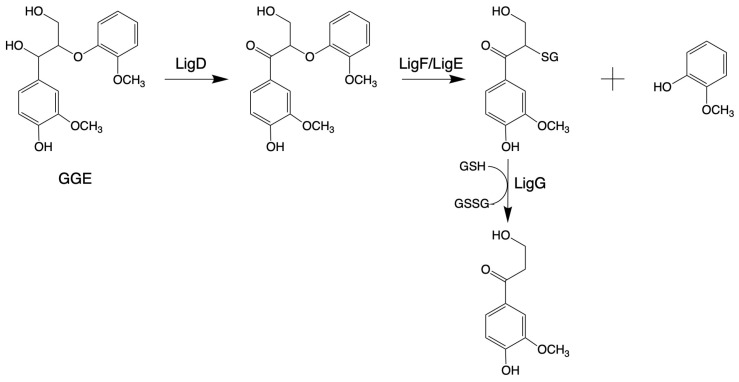
Pathways for the cleavage of *β*-*O*-4 bond by *β*-Etherase in *Sphingobium* sp. SYK-6 (adapted from [34]).

**Figure 3 microorganisms-13-00935-f003:**
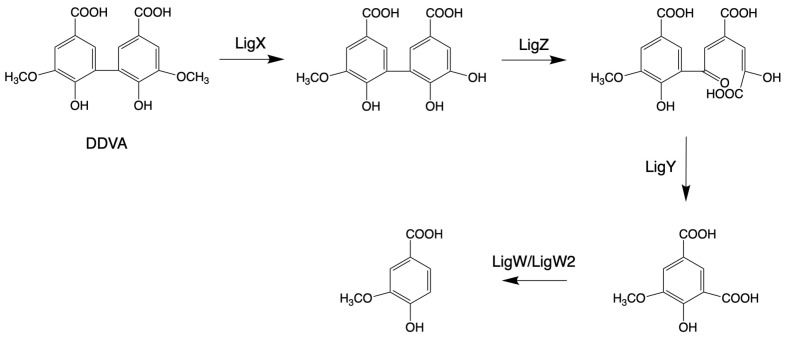
Pathways for the structural cleavage of biphenyl moieties in *Sphingobium* sp. SYK-6 (adapted from [34]).

**Figure 4 microorganisms-13-00935-f004:**
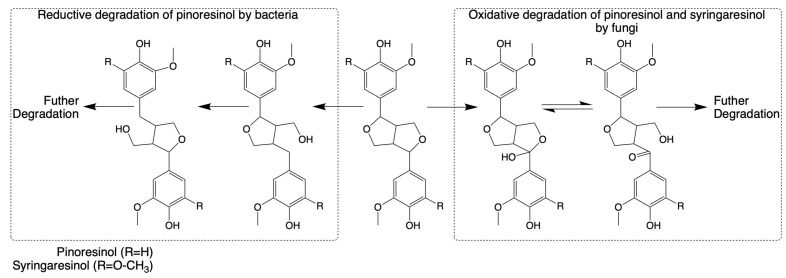
Pathways for the cleavage and subsequent degradation of pinoresinol and syringaresinol by bacteria and fungi, respectively (adapted from [34,164]).

**Figure 5 microorganisms-13-00935-f005:**
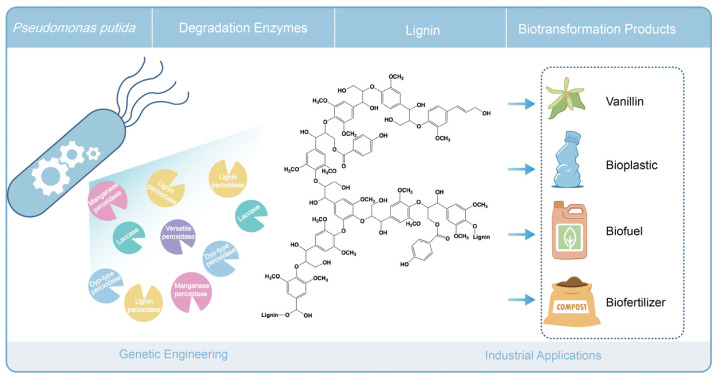
Conceptual representation of lignin biodegradation and engineering in *Pseudomonas putida*.

**Table 3 microorganisms-13-00935-t003:** Enzymes capable of cleaving specific lignin linkages. This table summarizes the types of enzymes involved in cleaving specific lignin linkages, their classification, their names, their source strains, the substrates utilized, and the relevant references. The enzymes listed are crucial for understanding the biochemical pathways of lignin degradation and highlight the diverse microbial sources capable of lignin bioconversion.

Linkage	Type of Enzyme	Name	Strains Source	Substrate Intracellular	Location	Refs.
*β*-*O*-4	*β*-Etherase	LigE; LigF	*Sphingobium* sp. SYK-6; *Novosphingobium* sp. *strain* PP1Y	1-(4-hydroxy-3-methoxyphenyl)-2-(2-methoxyphenoxy)-1,3-propanediol	Intracellular	[5,116]
		Ds-GST1	*Dichomitus squalens* LYAD-421 SS1	dimeric lignin model compound contains β-O-4 aryl ether bond	Intracellular	[117]
	laccase-like multicopper oxidases	CopA	*Pseudomonas putida* KT2440; *Pseudomonas fluorescens* Pf-5	guaiacylglycerol-β-guaiacyl ether (GGE)	Secreted	[70]
	Dye-decolorizing peroxidase	Rh_DypB	*Rhodococcus jostii* RHA1	GGE	Secreted	[118]
	heme-containing ligninolytic peroxidase	Versatile peroxidase	*Physisporinus vitreus*	guaiacylglycerol β-guaiacyl ether (β-*O*-4 dimer)	Secreted	[119]
	heme-containing peroxidases	Manganese peroxidase	*Phanerochaete chrysosporium*	1-(3,5-dimethoxy-4-hydroxyphenyl)-2-[4-(hydroxymethyl)-2-methoxyphenoxy]-1,3-dihydro-xypropane	Secreted	[120]
	Laccases	Small laccase (SLAC)	*Streptomyces*	LM-OH (a phenolic *β*-*O*-4 lignin model compound)	Secreted	[86]
		Laccase	*Bacillus ligniniphilus* L1	alkaline lignin and milled wood lignin	Intracellular	[121]
5-5	C-C hydrolase	LigY	*Sphingomonas paucimobilis* SYK-6	2,2′-dihydroxy-3,3′-dimethoxy-5,5′-dicarboxybiphenyl (DDVA)	Intracellular	[122,123,124]
	heme-containing ligninolytic peroxidase	Versatile peroxidase	*Physisporinus vitreus*	dehydrodivanillic alcohol (5-5′ dimer)	Secreted	[119]
	laccase-like multicopper oxidases	CopA	*Pseudomonas putida* KT2440; *Pseudomonas fluorescens* Pf-5	DDVA	Secreted	[70]
β-β	phenol-oxidizing enzymes		*Fusarium solani* M-13-1	l-syringaresinol	Secreted	[125]
Other bonds	oxygen oxidoreductase	Laccase	*Staphylococcus arlettae* S1-20		Secreted	[126]
		Lignin peroxidase	*Streptomyces viridosporus* T7A	GGE	Secreted	[127]
		Lignin peroxidase	*Trametes versicolor* IBL-04	veratryl alcohol	Secreted	[128]

**Table 4 microorganisms-13-00935-t004:** Typical genomic tools for cloning, insertions, and deletions in *P. putida*.

Genomic Tools	Purpose	References
Tn5-based transposon system	Random insertion of genes	[56]
Tn7-based transposon system	Insertions	[182]
Inducible expression systems: XylS/Pm, LacIQ/Ptrc, Plac, Ptac	Expression of target genes	[56,183,184]
pEMG	Scarless deletions and insertions	[185]
pSNW	Scarless deletions and insertions	[186]
CRISPR-Cas9 systems	Precise genome editing, allowing for targeted gene knockouts and insertions	[187,188]
CRISPR-Cas3 systems	Precise genome editing, allowing for targeted gene knockouts and insertions	[189]
phi15-based expression system	Expression of target genes	[190]

## Data Availability

This review does not present any new data.

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
