# Peer review of "Lignin-Degrading Enzymes and the Potential of Pseudomonas putida as a Cell Factory for Lignin Degradation and Valorization"

_microorganisms, 2025, doi:10.3390/microorganisms13040935_

Round 1

Reviewer 1 Report

Comments and Suggestions for Authors As a general comment, I recommend that the authors note somewhere in the manuscript that although bacteria have a great metabolic potential to convert small lignin-like molecules into valuable products, their extracellular enzymes are not as efficient in depolymerizing lignin as those of white rot fungi. Therefore, the hybrid system [depolymerization by enzymes of white rot fungi] -> [bacterial conversion] is now a common concept that many laboratories are trying to develop. In particular, note that: (1) Lines 77-78: It is highly debatable that "bacterial lignin degradation is more specific than fungal lignin degradation". I recommend refraining from making such general, unsubstantiated statements. Moreover, in lines 81-83 you emphasize that there is little information on this topic. (2) For all enzymes mentioned in this manuscript, please clearly indicate whether they are secreted or intracellular. Intracellular enzymes can only theoretically participate in lignin degradation as recombinant enzymes. However, in nature, intracellular enzymes can only work on small molecules, that structurally resemble parts of lignin. (3) In Figure 5, some enzyme names are illegible. Please improve it.

Reviewer 2 Report

Comments and Suggestions for Authors

This review focuses on the biotechnological valorization of lignin in general and the potential of Pseudomonas putida in this process in particular. In my opinion, this review is systematic and comprehensive. At the beginning it provides all the necessary background information on the topic of lignin-degrading microorganisms and enzymes and then focuses on P. putida. I think this paper definitely deserves publication; however, I have a few small comments that, if corrected, would improve the manuscript:

   -> Lines 73-75: Factual error! Brown-rot fungi cannot degrade lignin; moreover, this group of fungi does not have lignin-degrading peroxidases in their genomes, and the role of laccases in brown-rots is still unknown. Brown-rot fungi can only slightly modify lignin by oxidizing its phenolic units.

   -> Lines 114-115: Please add Trametes hirsuta and Steccherinum ochraceum to this list of current model lignin-degrading fungi. Seven laccases have been isolated and characterized (importantly with regard to their glycosylation patterns) from different S. ochraceum strains, and vast literature is available on the laccases of T. hirsuta.

-> It is worth noting regarding enzymatic systems of white-rot fungi (especially in section 3.1.1) that a link between the lignin molecular profile and the fungal exoproteome has been demonstrated for T. hirsuta. It is also worth noting that Peniophora lycii LE-BIN 2142 has been shown completely lacking ligninolytic peroxidases, a well-known marker of white rot fungi in its exoproteome; however, its exoproteome contained several laccase isozymes and previously uncharacterized FAD-binding domain-containing proteins.

To sum up, this is a good paper! I recommend it for publication after minor revision.
